# Integrated Fiber-FSO WDM Access System with Fiber Fault Protection

**Chien-Hung Yeh** [1,*] , **Wen-Piao Lin** [2], **Syu-Yang Jiang** [1], **Shang-En Hsieh** [1], **Ching-Hsuan Hsu** [1] **and Chi-Wai Chow** [3]

[1] Department of Photonics, Feng Chia University, Taichung 407802, Taiwan; m1011162@o365.fcu.edu.tw (S.-Y.J.); m1008756@o365.fcu.edu.tw (S.-E.H.); m1011305@o365.fcu.edu.tw (C.-H.H.)

[2] Department of Electrical Engineering, Chang Gung University, Taoyuan 333323, Taiwan; wplin@mail.cgu.edu.tw

[3] Department of Photonics, National Yang Ming Chiao Tung University, Hsinchu 300093, Taiwan; cwchow@nycu.edu.tw

\* Correspondence: yehch@fcu.edu.tw

**Abstract:** In this study, we propose a new wavelength-division-multiplexing passive optical network (WDM-PON) system to support the blended fiber-free space optics (FSO) signal access. To provide the fiber and FSO traffics simultaneously, the C- and L-band channels are applied in the presented PON, respectively. Moreover, to avoid the fiber breakpoint in the fiber access traffic, the proposed WDM access architecture also can provide the self-restored mechanism by applying simple fiber routing path. In addition, the corresponding signal performances of fiber and FSO channels are also executed experimentally for demonstration.

**Keywords:** wavelength-division-multiplexing (WDM); fault protection; passive optical network (PON); free space optics (FSO)

## 1. Introduction

Recently, with the increase in the access data traffic numerously owing to the speedy progress of cloud service, quantum computing, artificial intelligence (AI), data center, big data, 4K/8K video, online game, and 3D holographic application, the network service providers need to provide the wide network capacity, low latency, high-speed rate, and cost-effective deployment for end user [1–3]. Thus, the passive optical network (PON) would be the best candidate for the last mile data traffic access [4,5]. Here, the time-division-multiplexing (TDM) and wavelength-division-multiplexing (WDM) access are currently the most used technologies. The TDM- and WDM-PONs are the point to multi-point (PTMP) and point to point (PTP) access system [6,7], respectively. The relative development and standard TDM-PON systems have also been demonstrated fully, such as the Ethernet-PON (EPON, IEEE 802.3ah), Gigabit-PON (GPON, ITU-T G.984), next-generation PON (NG-PON1, ITU-T G.987) and NG-PON2 (ITU-T G.989) [8–11]. The traffic rates of these TDM-PONs were from 1 to 40 Gb/s for broadcasting and power-sharing downstream data due to the PTMP operation. To achieve higher data capacity and network security, the PTP-based WDM-PON systems with the traffic rate from 1 to 60 Gb/s and even to 100 Gb/s have been studied and demonstrated extensively [12–15]. Due to the high amount of data transmitted in WDM system, the stability and reliability of the fiber access network is extremely important [16]. Therefore, under high-speed and wide capacity signal transmission in such PON networks, if the connected fiber path is broken, it will affect the network connection of the related end-users. To prevent the occurrence of fiber breakpoint, the self-healing operation in WDM-PON network is also an important research topic [17–20]. However, the previous works needed to add extra more passive and active devices in the PON architecture to avoid the fiber fault occurrence.

Furthermore, in some situations, the conventional PON might not be suitable for the installation of fiber owing to the geographical restrictions [21]. In PON network, the fiber can be replaced by using the free space optics (FSO) communication in some parts of the optical distribution network (ODN), where installing fiber is hard and expensive. Therefore, use of the hybrid free space optics (FSO) communication and PON access have also been proposed and proven [21–23]. In addition to being used in environments where optical fibers cannot be deployed, the FSO transmission also makes the access network more flexible and reliable [24].

In this paper, an integrated fiber-FSO WDM-PON system with self-protected mechanism against fiber fault is proposed. 10 Gb/s on-off keying (OOK) bidirectional WDM fiber and FSO signals in the same PON architecture are achieved simultaneously. Here, we apply C- and L-band wavelengths regarding as the fiber and FSO data traffics, respectively. To demonstrate the signal performances of fiber and FSO access, the eight C- and L-band wavelengths with 0.8 nm channel spacing of 1535.04 to 1540.56 nm ($\lambda_{1C}$ to $\lambda_{8C}$) and 1560.61 to 1566.31 nm ($\lambda_{1L}$ to $\lambda_{8L}$) are exploited for presentation, respectively. The access point (AP) can be deployed independently or integrated into optical network unit (ONU) to access the FSO signal according to the actual environment and demand. The proposed WDM-PON system not only can deliver the fiber and FSO signals simultaneously, but also can provide the fault protection for guaranteeing fiber signal access. All the obtained power budgets of testing signals can meet under the forward error correction (FEC) level in the presented PON architecture. Moreover, all the redundant power budget of FSO traffic also can support bidirectional free space transmission of 800 m without signal amplification based on the previous optical design system [25]. Compared with the related recent study [24], the proposed self-protected WDM-PON architecture can not only support the fiber and FSO transmissions at the same time, but also provide self-healing fiber breakpoint protection with simple and cost-effective implement.

## 2. Proposed PON Access Architecture

The original WDM-PON architecture is shown in Figure 1a. Each WDM transceiver (TRx) is placed in the central office (CO) to transmit the corresponding downstream traffic and detect the upstream signal through the $1 \times N$ arrayed waveguide grating (AWG) device for WDM access, respectively. Each connected port of AWG has 3 dB bandwidth of 0.4 nm. In addition, the related upstream traffic of the optical network unit (ONU) can transmit via the same fiber path. As we know, the feeder and distribution fibers are between the CO and remote node (RN) and RN and ONUs in PON system, respectively. Furthermore, to deliver the FSO signal by the current WDM-PON architecture, the access points (APs) and optical wireless units (OWUs) are needed to build for FSO connection [24,25], as illustrated in Figure 1b. Hence, the FSO traffic can be integrated in PON for optical wireless data access.

To provide the fiber and FSO data links in WDM access network simultaneously, a simple PON architecture is proposed as shown in Figure 2. In the CO, the C- and L-band WDM laser sources can be applied to regard as the baseband and FSO signals, respectively. Thus, all the WDM TRx of C- (TRx$_{1C}$, TRx$_{2C}$, . . . and TRx$_{NC}$) and L-bands (TRx$_{1L}$, TRx$_{2L}$, . . . and TRx$_{NL}$) are connected to the corresponding port of the $1 \times N$ arrayed waveguide grating (AWG$_1$ and AWG$_2$) for the downstream fiber and FSO wavelength connections, respectively. In addition, the corresponding TRx$_{(N+1)C}$, TRx$_{(N+2)C}$, . . . and TRx$_{(2N)C}$, and TRx$_{(N+1)L}$, TRx$_{(N+2)L}$, . . . and TRx$_{(2N)L}$ are placed in the corresponding ONU and OWU for fiber and FSO upstream traffics, respectively.

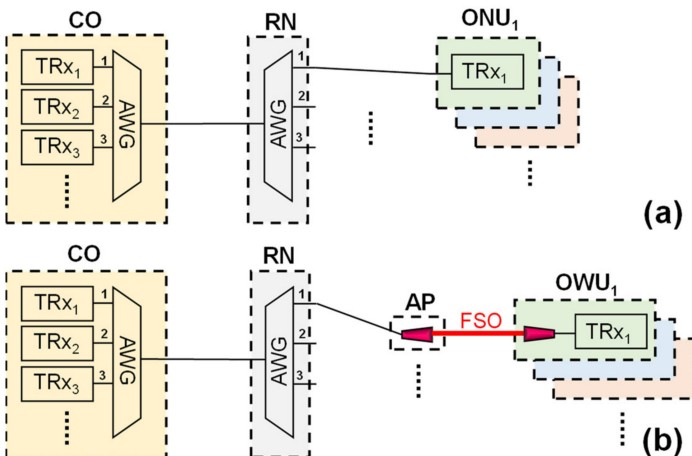

**Figure 1.** (**a**) Conventional WDM-PON system. (**b**) FSO transmission system integrated with WDM-PON.

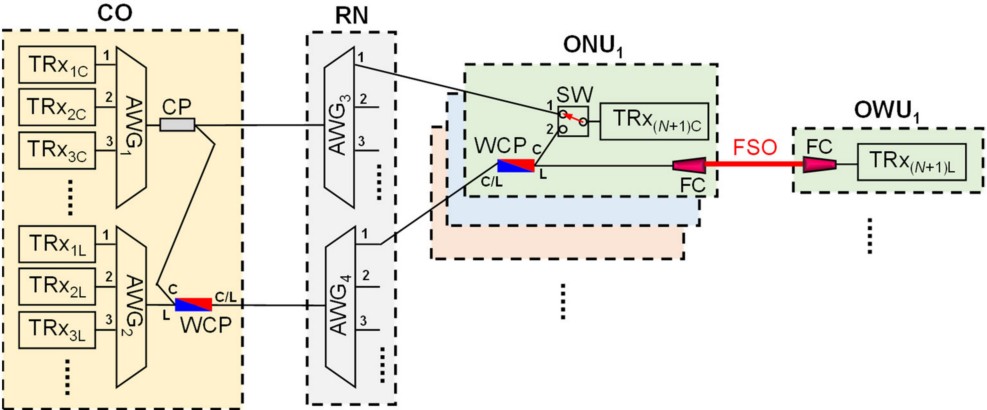

**Figure 2.** Proposed self-protected WDM-PON system together with fiber and FSO signals for data access against fiber fault.

However, in the PON access, the fiber fault also an important issue for high-speed fiber data link. Thus, the self-protected mechanism against fiber breakpoint will be significant consideration. In this investigation, to provide the fault protection of C-band fiber signal link simultaneously, the wideband $1 \times 2$ and 50:50 optical coupler (CP) and C/L band WDM coupler (WCP) are also required to exploit at the output port of $AWG_1$ and $AWG_2$, respectively, as seen in Figure 2. The corresponding downstream fiber ($\lambda_{1C}$, $\lambda_{2C}$, ... and $\lambda_{NC}$) and FSO signals ($\lambda_{1L}$, $\lambda_{2L}$, ... and $\lambda_{NL}$) will be separated at the RN by the $AWG_3$ and $AWG_4$, respectively, and then enter the same corresponding ONU, as illustrated in Figure 2. Here, the $AWG_1$ to $AWG_4$ are the same components in the proposed PON network. A subscript font from 1 to 4 is used to distinguish where the AWG is placed. As we know, due to the periodic characteristic of AWG, the downstream wavelengths of $\lambda_{1C}$ and $\lambda_{1L}$, $\lambda_{2C}$ and $\lambda_{2L}$, ... and $\lambda_{NC}$ and $\lambda_{NL}$ can enter the connected port of "1", "2", ... and "$N$", respectively. Hence, the downstream fiber and FSO signals of $\lambda_{1C}$ and $\lambda_{1L}$, $\lambda_{2C}$ and $\lambda_{2L}$, ... and $\lambda_{NC}$ and $\lambda_{NL}$ are sent from the CO via the $AWG_1$, $AWG_2$, $AWG_3$, and $AWG_4$, respectively, and arrive the corresponding $ONU_1$, $ONU_2$, ... and $ONU_N$. Similarly, the upstream fiber and FSO signals of $\lambda_{(N+1)C}$ and $\lambda_{(N+1)L}$, $\lambda_{(N+2)C}$ and $\lambda_{(N+2)L}$, ... and $\lambda_{(2N)C}$ and $\lambda_{(2N)L}$ are also through the same connected ports for data transmission via the AWGs, respectively, as seen in Figure 2.

As plotted in Figure 2, the $ONU_1$ includes a WCP, a $1 \times 2$ optical switch (SW), a fiber collimator (FC) and a $TRx_{(N+1)C}$. The WCP of $ONU_1$ is applied to separate the downstream fiber ($\lambda_{1C}$) and FSO wavelengths ($\lambda_{1L}$). The FC is applied to generate the FSO signal

for wireless transmission. Initially, the SW is connected to the "1" point for fiber signal connection. As seen in Figure 2, the $\lambda_{1C}$ signal of the CO will be separated two wavelengths by the $1 \times 2$ CP passing through the upper (fiber path) and lower connecting fibers (FSO path) simultaneously and then into the ONU$_1$. Moreover, the separated $\lambda_{1C}$ signal from the lower connecting fiber would be blocked by the SW to avoid the signal interference in the ONU$_1$, as exhibited in Figure 2. Therefore, the fiber signal $\lambda_{1C}$ will be received by the photodiode (PD) of TRx$_{(N+1)C}$ after passing through the SW.

For the L-band FSO transmission, the FC of AP can be combined into the ONU$_1$ for converting the optical wireless signal, as exhibited in Figure 2. Here, the FC is exploited to deliver the corresponding FSO $\lambda_{1L}$ from the CO and then enter the OWU$_1$ through the lower connecting fiber, as shown in Figure 2. Although the C-band baseband signals will also be transmitted from the lower connecting fiber, but it will be blocked by the WCP of ONU$_1$. This means that the C-band fiber signal cannot affect the L-band FSO signal. Through a wireless FSO transmission length, the $\lambda_{1L}$ signal is collected by the FC of the OWU$_1$ and then into the PD for decoding. In the demonstration, the upstream fiber traffic of ONU$_1$ and the FSO signal of OWU$_1$ can also go back to the CO through the same transmission path, respectively, as seen in Figure 2. As a result, the proposed WDM-PON system not only can provide the fiber access traffic, but also can support the wireless FSO signal via the same ONU.

As mentioned above, the downstream wavelengths of $\lambda_{1C}$, $\lambda_{2C}$, ... and $\lambda_{NC}$ and $\lambda_{1L}$, $\lambda_{2L}$, ... and $\lambda_{NL}$ are connected via the upper and lower connecting fibers into the corresponding ONU and OWU in normal state, respectively, as shown in Figure 2. In addition, the upstream fiber and FSO signals of $\lambda_{(N+1)C}$, $\lambda_{(N+2)C}$, ... and $\lambda_{(2N)C}$, and $\lambda_{(N+1)L}$, $\lambda_{(N+2)L}$, ... and $\lambda_{(2N)L}$ are also through the same fiber paths back to the CO. In this demonstration, the designed PON network also can provide the self-protected function against fiber fault for C-band fiber signal link. In general, the fiber breakpoint may be occurred on the feeder fiber (between CO and RN) and distribution fiber (between RN and ONU$_N$) to interrupt the signal connection. For example, suppose there is a fiber breakpoint at the "F1" point between RN and ONU$_1$, as seen in Figure 3a. The bidirectional fiber signal connection between the CO and ONU$_1$ will interrupt at this moment. To prevent the signal disconnection, the SW will switch to the "2" position for routing signal traffic through the lower connecting fiber, when the ONU$_1$ cannot receive the corresponding downstream signal from the CO, as illustrated in Figure 3a. Hence, the downstream and upstream fiber traffics will pass through the lower fiber path for signal reconnection. Once the fiber breakpoint is repaired, the SW switches back to the "1" position for normal connection. Therefore, the presented WDM-PON architecture can not only provide the fiber and FSO traffics simultaneously, but also can support the fiber fault protection in the fiber PON connection to provide the seamless signal access.

As mentioned above, the corresponding AP can be also detached from ONU for FSO connection in the proposed WDM-PON system, as illustrated in Figure 4. The ONU and AP is linked by a short length of fiber path. If a fiber breakpoint is happened at the "F1" or "F2" points, the fiber access data can be routed through the FSO link path (lower connecting fiber) for fault protection. The ONU, AP and OWU modules of Figures 2 and 4 are similar and can be exploited in the proposed self-protected PON network simultaneously according to the actual field environment and requirement for deployment.

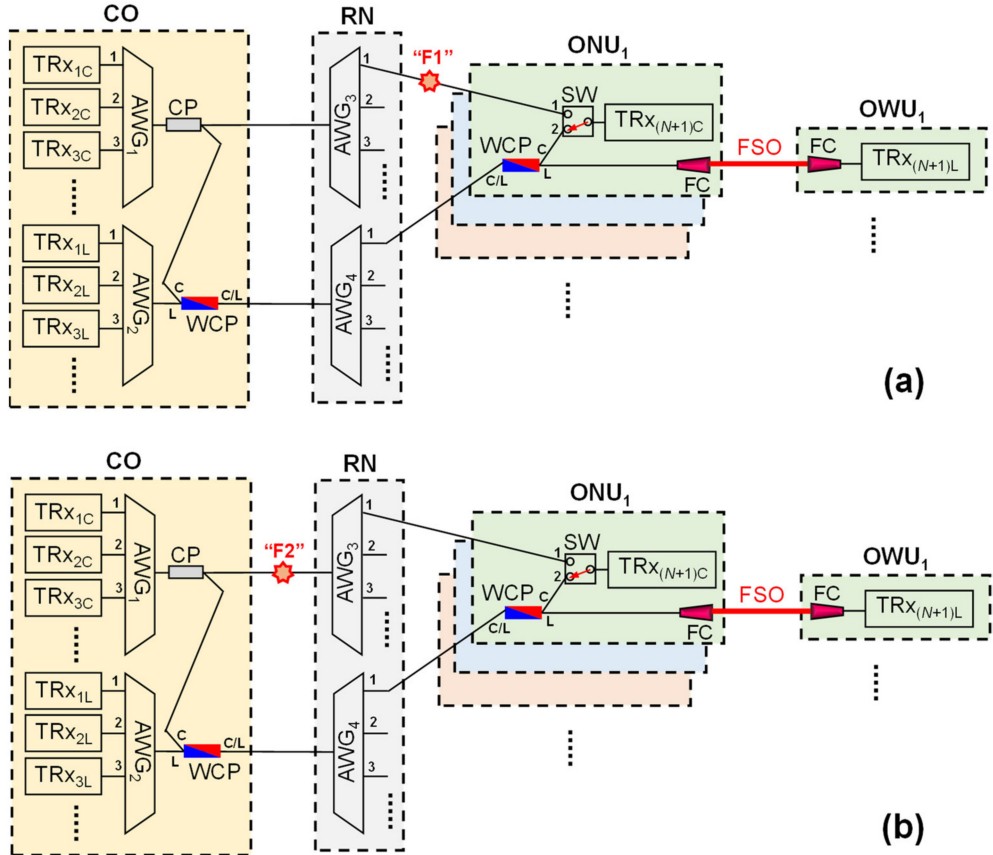

**Figure 3.** The occurrence of fiber fault at the (**a**) "F1" and (**b**) "F2" points in the presented WDM-PON system, respectively.

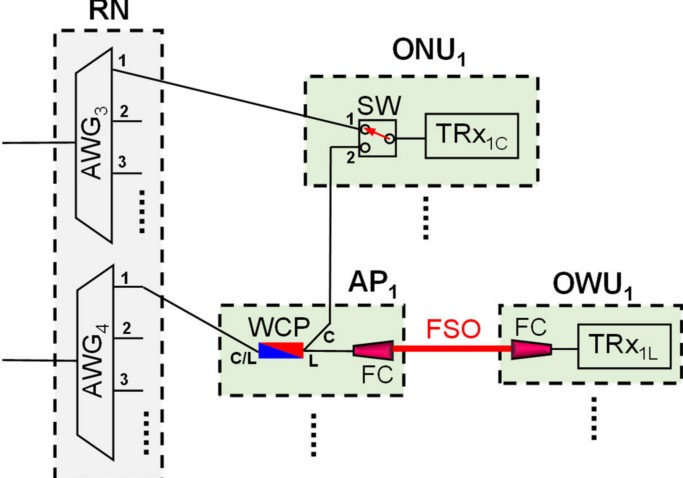

**Figure 4.** The new designed architectures of RN, ONU, AP and OWU in the hybrid fiber-FSO WDM-PON network.

## 3. Experiment and Results

In the experiment, we use a C-band tunable laser source (TLS) to provide the different output wavelengths for the downstream ($\lambda_{1C}$ to $\lambda_{4C}$) and upstream fiber links ($\lambda_{5C}$ to $\lambda_{8C}$), as displayed in Figure 5a. The TLS (General Photonics, TLS-101-C) has ITU-T specification tuning range from 1528.77 to 1563.86 nm with 13 dBm largest output power and 0.4 nm tuning step. Each WDM wavelength connects to the polarization controller (PC) and 10 GHz Mach-Zehnder modulator (MZM) to carry 10 Gb/s on-off keying (OOK) modulation

signal for regarding as the symmetric downstream and upstream traffics in the CO, ONU, respectively. The PC can control the polarization state and obtain the optimal output power. Two 3-port optical circulators ($OC_1$ and $OC_2$) are applied to isolate the downstream and upstream transmissions. The modulated signal will travel through a length of 25 km single-mode fiber (SMF) for bidirectional data connection. Furthermore, to enhance the modulation signal performance, an optical pre-amplifier is placed in front of the 10 GHz PIN photodiode (PD) for measurement.

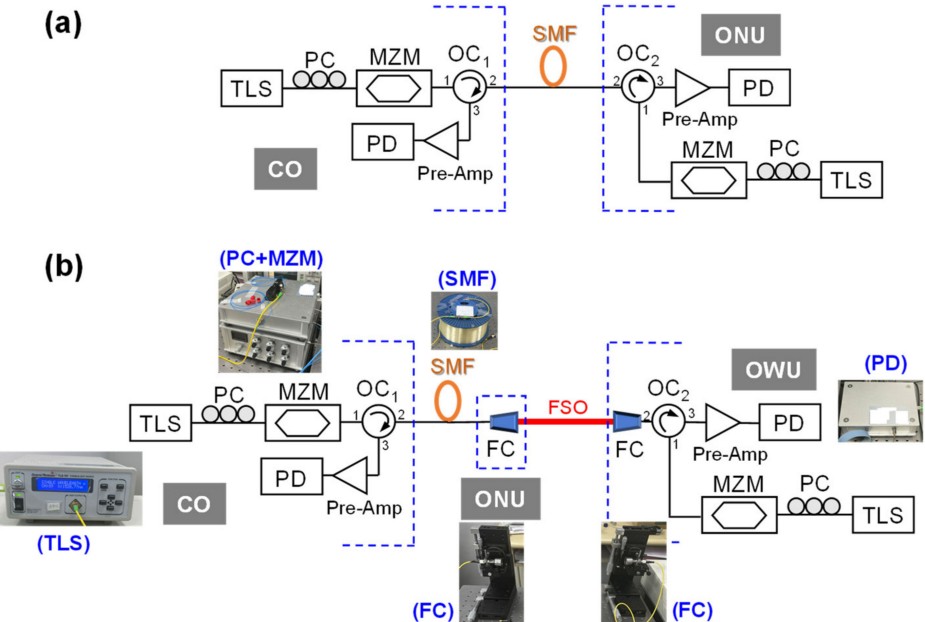

**Figure 5.** Experimental setup of (**a**) fiber and (**b**) FSO transmissions, respectively. The photos of insets are the corresponding active and passive components.

To demonstrate the FSO transmission in PON access, we can use a new setup for signal measurement, as plotted in Figure 5b. Here, we also apply 10 Gb/s OOK on the MZM for the wireless FSO traffic. To generate and collect the FSO signal, two FCs are exploited between the ONU and OWU. In the experiment, the related parameters of diameter, divergence angle and focal length of FC are 2 cm, 0.016° and 3.713 cm, respectively. In addition, the acceptable wavelength bandwidth of FC is from 1050 to 1620 nm. The wireless FSO length is ~2 m between OWU and ONU for the convenience of measurement. Moreover, the SMF link is also set at 25 km for data transmission. Through the optimal alignment of FSO link, the insertion loss between two FCs is around 3 dB here.

Next, to realize the 10 Gb/s OOK bit error rate (BER) performance, eight WDM wavelengths of 1535.04 ($\lambda_{1C}$), 1535.82 ($\lambda_{2C}$), 1536.61 ($\lambda_{3C}$), 1537.40 ($\lambda_{4C}$), 1538.19 ($\lambda_{5C}$), 1538.98 ($\lambda_{6C}$), 1539.77 ($\lambda_{7C}$), and 1540.56 nm ($\lambda_{8C}$) are selected as the downstream ($\lambda_{1C}$ to $\lambda_{4C}$) and upstream fiber traffics ($\lambda_{5C}$ to $\lambda_{8C}$), respectively. Based on the periodic feature of AWG, the wavelengths of $\lambda_{1C}$ and $\lambda_{5C}$, $\lambda_{2C}$ and $\lambda_{6C}$, $\lambda_{3C}$ and $\lambda_{7C}$, and $\lambda_{4C}$ and $\lambda_{8C}$ can be through the connected ports of 1 to 4 for bidirectional signal link, respectively. In the experiment, the output power of TLS at the port "2" of $OC_1$ and $OC_2$ is set at 7.5 dBm to carry the OOK modulation signal to avoid the nonlinear effect, as illustrated in Figure 5a. Here, the measured 10 Gb/s OOK BER performances of the wavelengths of $\lambda_{1C}$ to $\lambda_{8C}$ at the back-to-back (BTB) state are observed in Figure 6a, respectively. The measured power sensitivities of $\lambda_{1C}$ to $\lambda_{8C}$ are −29.5, −28, −27, −26, −25.5, −25.5, −25 and −25 dBm at the forward error correction (FEC) level (BER $\leq 3.8 \times 10^{-3}$), respectively. Hence, the achievable power budgets of $\lambda_{1C}$ to $\lambda_{8C}$ are 36.5, 35.5, 34.5, 33.5, 33, 33, 32.5 and 32.5 dB, respectively. Through 25 km SMF transmission, the 10 Gb/s BER measurements of eight wavelengths are exhibited in Figure 6b. The detected power sensitivities of $\lambda_{1C}$ to $\lambda_{8C}$ are −30.5, −29,

−28, −27.5, −27, −26.5, −26.5 and −26.5 dBm at the FEC target, respectively. In addition, the obtained power sensitivity at the FEC can be improved from 1 to 1.5 dB after through 25 km SMF link, as seen in Figure 6a,b. This is because we exploit the −0.7 chirp parameter MZM in the proposed PON to pre-compensate the fiber dispersion. The power budget of 35 to 38 dB can be achieved after 25 km SMF link. The insets (i) to (iv) of Figure 6a,b are the corresponding eye diagrams of 1535.04 nm ($\lambda_{1C}$) at the received powers of −29.5, −23, −30.5 and −24 dBm, respectively. The measured eyes of insets (ii) and (iv) are open and clear. In addition, the insets (i) and (iii) also can be distinguished for observation.

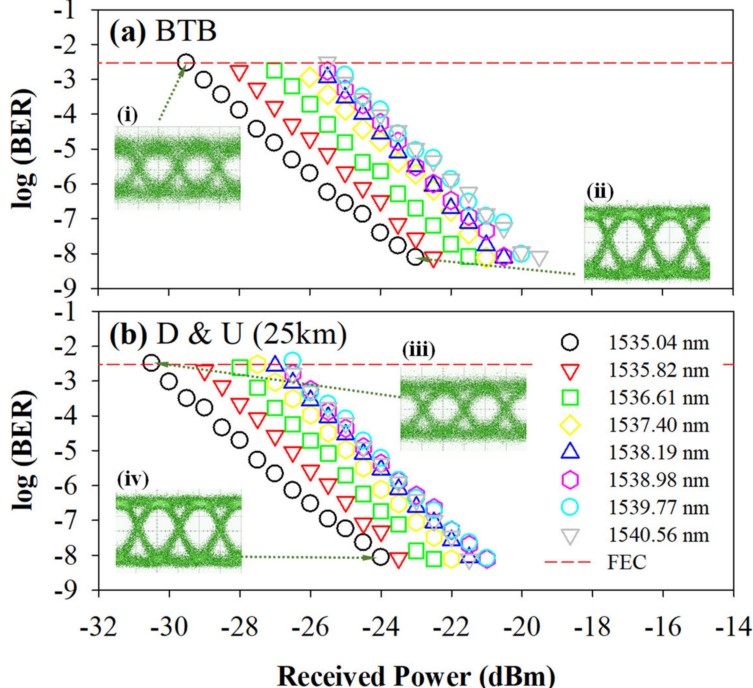

**Figure 6.** Measured BER performances of $\lambda_{1C}$ to $\lambda_{8C}$ (**a**) at the BTB state and (**b**) through 25 km SMF, respectively. Insets (i) to (iv) are the observed corresponding eye diagrams under different received powers.

Based on the presented PON architecture of Figure 2 and experimental setup of Figure 5a, the downstream signal will pass through the $AWG_3$ (6 dB), SW (1 dB), 25 km fiber (0.2 × 25 = 5 dB) and $OC_2$ (0.5 dB) after leaving the $CO_1$ in normal state, respectively. Thus, 17 dB total insertion loss of each downstream traffic is produced by the proposed PON system. The upstream traffic also passes through the SW, $AWG_3$, and 25 km SMF. The upstream signal also results in the same insertion loss of 17 dB. Commonly, an optical amplifier is applied in the CO to compensate the insertion loss of upstream signal. Therefore, the insertion loss induced by the $AWG_1$, $AWG_2$, CP and WCP can be ignored in the CO site.

Once the fault is happened at the "F1" or "F2" points of Figure 3a or Figure 3b, respectively, the downstream and upstream traffics will transmit through the SW (1 dB), WCP (1 dB), $AWG_4$ (6 dB), 25 km SMF (0.2 × 25 = 5 dB) and OC (0.5 dB), when the SW switches to the point "2". Thus, the total insertion loss of downstream and upstream signals is both 13.5 dB, while the protection mechanism of PON system is operated. Moreover, as mentioned above, through 25 km fiber connection, the corresponding power budgets of downstream and upstream traffics can be achieved between 35 and 38 dB and 34 and 34.5 dB, respectively. As a result, the achieved power budgets can meet with the total insertion loss induced by the fiber access network under the normal and fault protection states.

In the FSO measurement, the selected L-band wavelengths of 1560.61 ($\lambda_{1L}$), 1561.42 ($\lambda_{2L}$), 1562.23 ($\lambda_{3L}$) and 1563.05 nm ($\lambda_{4L}$); and 1563.86 ($\lambda_{5L}$), 1564.68 ($\lambda_{6L}$), 1565.50 ($\lambda_{7L}$) and 1566.31 nm ($\lambda_{8L}$) are served as the downstream and upstream traffics, respectively. Here, the

output power of TLS at the port "2" of $OC_1$ and $OC_2$ is 7.5 dBm, as illustrated in Figure 5b. We also apply 10 Gb/s OOK format on MZM to generate the symmetric downstream and upstream FSO signals in the presented PON system. Figure 7a,b exhibit the measured 10 Gb/s OOK FSO BERs of downstream ($\lambda_{1L}$ to $\lambda_{4L}$) and upstream transmissions ($\lambda_{5L}$ to $\lambda_{8L}$) after 25 km SMF link and 2 m free space connection, respectively. The observed power sensitivities of eight wavelengths ($\lambda_{1L}$ to $\lambda_{8L}$) are $-21.5$, $-21.5$, $-21.5$ and $-21.5$ dBm and $-22$, $-22$, $-22$ and $-22$ dBm below the FEC threshold, respectively. Thus, the achieved power budgets of downstream and upstream signals are 29 and 29.5 dB, respectively. The measured sensitivity of L-band FSO wavelength is inferior to that of the C-band signal, as shown in Figures 5 and 6, due to fiber dispersion effect at the longer wavelengths. Here, the insets (i) to (iv) of Figure 7a,b are the measured eye diagrams of 1560.61 ($\lambda_{1L}$) and 1563.86 nm ($\lambda_{5L}$) at the received powers of $-21.5$ and $-15$ dBm and $-22$ and $-14.5$ dBm, respectively. These measured eyes also seem clear below the FEC limit.

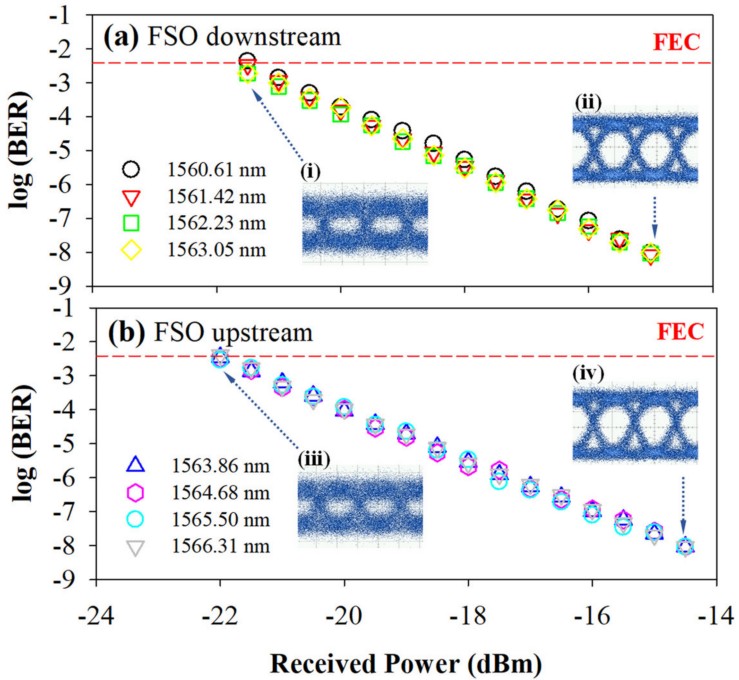

**Figure 7.** Measured FSO BER performances of (**a**) $\lambda_{1L}$ to $\lambda_{4L}$ and (**b**) $\lambda_{5L}$ to $\lambda_{8L}$ through 25 km SMF and 2 m FSO links, respectively. Insets (i) to (iv) are the observed corresponding eye diagrams under different received powers.

Furthermore, the FSO downstream and upstream wavelengths will induce the same total loss of 15.5 dB, including the $AWG_4$ (6 dB), WCP (1 dB), 25 km SMF ($0.2 \times 25 = 5$ dB), OC (0.5 dB) and two FCs (3 dB), respectively. Here, the redundant budgets of 13.5 and 14 dB can be achieved to determine the wireless lengths of the downstream and upstream FSO transmissions, respectively. According to the previous FSO study [25], the obtained power budget of 13.5 dB can support 800 m free space transmission link based on the prior designed optical system without signal amplification. If we want to extend the FSO transmission distance, an optical amplifier can be added in the ONU side for signal amplification. To improve the signal performances of fiber and FSO traffics, the simplified PON architecture can improve the power budget for providing a longer optical fiber and FSO transmission distance. In addition, to clearly express the signal performance obtained in the PON system, the corresponding results of fiber and FSO signal is listed in Table 1.

**Table 1.** The corresponding results of fiber and FSO signal performances.

| Characteristic | Fiber Traffic | FSO Traffic |
|---|---|---|
| Insertion Loss of Downstream (dB) | 17 | 15.5 |
| Insertion Loss of Upstream (dB) | 17 | 15.5 |
| Downstream Sensitivity (dBm) | 29.5 to 26 | 21.5 |
| Upstream Sensitivity (dBm) | 25.5 to 25 | 22 |
| Power Budget of Downstream (dB) | 35 to 38 | 29 |
| Power Budget of Upstream (dB) | 34 to 34.5 | 29.5 |
| FSO Link Length (m) | / | 800 |

## 4. Conclusions

In summary, we proposed a hybrid fiber-FSO WDM-PON architecture with self-protected operation against fiber breakpoint. Here, the C- and L-band wavelengths were applied to regard as the 10 Gb/s OOK fiber and FSO signals in the presented WDM-PON network. In the demonstration, eight C- and L-band WDM wavelengths were applied to act as the 10 Gb/s fiber and FSO signals for downstream ($\lambda_{1C}$, $\lambda_{2C}$, ... and $\lambda_{NC}$ and $\lambda_{1L}$, $\lambda_{2L}$, ... and $\lambda_{NL}$) and upstream traffics ($\lambda_{(N+1)C}$, $\lambda_{(N+2)C}$, ... and $\lambda_{(2N)C}$ and $\lambda_{(N+1)L}$, $\lambda_{(N+2)L}$, ... and $\lambda_{(2N)L}$), respectively. In the measurement, eight C-band fiber signals ($\lambda_{1C}$ to $\lambda_{8C}$) and eight L-band FSO signals ($\lambda_{1L}$ to $\lambda_{8L}$) were exploited, respectively. The measured power sensitivities of downstream and upstream fiber and FSO traffics were below $-27.5$ and $-26.5$ dBm and $-21.5$ and $-22$ dBm, after through 25 km SMF and 2 m wireless FSO transmissions, respectively. All the obtained power budgets of fiber and FSO signals would meet with the proposed WDM-PON network either in the normal state or under the self-healing protection at the FEC target. Therefore, the presented PON architecture not only supported the fiber access FSO signals, but also provided the fault protection for the WDM fiber access link. In addition, the proposed PON system was also easy to build and implement.

**Author Contributions:** Data curation, S.-Y.J., C.-H.H. and S.-E.H.; investigation, C.-H.Y.; supervision, W.-P.L. and C.-W.C.; writing—review and editing, C.-H.Y. All authors have read and agreed to the published version of the manuscript.

**Funding:** This paper was supported by Ministry of Science and Technology, Taiwan, under Grant MOST-110-2221-E-035-058-MY2.

**Conflicts of Interest:** The authors declare no conflict of interest.

## Abbreviations

The following abbreviations are used in this manuscript:

| | |
|---|---|
| AP | Access Point |
| AWG | Arrayed Waveguide Grating |
| BER | Bit Error Rate |
| CO | Central Office |
| CP | Optical Coupler |
| EPON | Ethernet-PON |
| FEC | Forward Error Correction |
| FSO | Free Space Optics |

| | |
|---|---|
| GPON | Gigabit-PON |
| MZM | Mach-Zehnder Modulator |
| NG-PON | Next Generation-PON |
| OC | Optical Circulator |
| OND | Optical Distribution Network |
| ONU | Optical Network Unit |
| OOK | On-Off Keying |
| OWU | Optical Wireless Unit |
| PC | Polarization Controller |
| PD | Photodiode |
| PON | Passive Optical Network |
| PTP | Point to Point |
| PTMP | Point to Multi-Point |
| RN | Remote Node |
| SMF | Single-Mode Fiber |
| SW | Optical Switch |
| TDM | Time-Division-Multiplexing |
| TLS | Tunable Laser Source |
| WCP | C/L Band WDM Coupler |
| WDM | Wavelength-Division-Multiplexing |

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
