# Peer review of "Integrated Fiber-FSO WDM Access System with Fiber Fault Protection"

_electronics, doi:10.3390/electronics11132101_

Round 1

Reviewer 1 Report

The authors have done some commendable work, however, following observations needs to be addressed.

1. On page-1, line 13, will you like to give full abbreviation of FSO?

2. One page 1-2, literature review is weak and some more credible references must be cited to give extensive outlook of the reported work. 

3. On page 2, line 62, why the sentence starts with a full stop?

4. On page-3, line 101, where is plot in the figure 2?

5. On page 2-3, figure 3 must be cited close to where discussed in literature.

6. Why is figure 3(b) mentioned later in the manuscript in line 210 page 6?

7. One Page 6, Line 195-196, What does mean by "All the measured eyes can be distinguished clearly". Are you reffering to the legend in figure 6, clarify that. 

8. Clarify your approach with a suitable application.

9. Also include list of abbreviations a the end.

10. You also have to compare it with a reported literature. 

11. Make a table to indicate all the results in a comparative manner. 

12. Also the abstract is weak, revise it properly if possible. 

Author Response

Reviewer #1:

The authors have done some commendable work, however, following observations needs to be addressed.

  1. On page-1, line 13, will you like to give full abbreviation of FSO?

[Response] Thanks to the reviewer’s useful suggestion. We have added the full abbreviation of FSO in the abstract as below.

…fiber-free space optics (FSO) signal access.

  1. One page 1-2, literature review is weak and some more credible references must be cited to give extensive outlook of the reported work.

[Response] We would like to thank the reviewer providing the suggestion. We have added more descriptions and references in the introduction section. Please see the revision.

The relative development and standard TDM-PON systems have also been demonstrated fully, such as the Ethernet-PON (EPON, IEEE 802.3ah), Gigabit-PON (GPON, ITU-T G.984), next-generation PON (NG-PON1, ITU-T G.987) and NG-PON2 (ITU-T G.989) [8-11]. The traffic rates of these TDM-PONs were from 1 to 40 Gb/s for broad-casting and power-sharing downstream data due to the PTMP operation.

To achieve higher data capacity and network security, the PTP-based WDM-PON systems with the traffic rate from 1 to 60 Gb/s and even to 100 Gb/s have been studied and demonstrated extensively [12–15].

Therefore, under high-speed and wide capacity signal transmission in such PON net-works, if the connected fiber path is broken, it will affect the network connection of the related end-users. To prevent the occurrence of fiber breakpoint, the self-healing operation in WDM-PON network is also an important research topic [17–20]. How-ever, the previous works needed to add extra more passive and active devices in the PON architecture to avoid the fiber fault occurrence.

Furthermore, in some situations, the conventional PON might not be suitable for the installation of fiber owing to the geographical restrictions [21]. In PON network, the fiber can be replaced by using the free space optics (FSO) communication in some parts of the optical distribution network (ODN), where installing fiber is hard and ex-pensive. Therefore, use of the hybrid free space optics (FSO) communication and PON access have also been proposed and proven [21–23].

The additional references are also included in the revision as below.

[8] Kazovsky, L. G.; Shaw, W.-T.; Gutierrez, D.; Cheng, N.; Wong, S.-W. Next-generation optical access networks. J. Lightw. Technol. 2007, 25, 3428–3442.

[9] Yeh, C.-H; Chow, C.-W.; Wang, C.-H.; Wu, Y.-F.; Shih, F.-Y.; Chi, S. Using OOK modulation for symmetric 40-Gb/s long-reach time-sharing passive optical networks. IEEE Photon. Technol. Lett. 2010, 22, 619–621.

[10] Chung, H. S.; Lee, H. H.; Kim, K. O.; Doo, K.-H.; Ra, Y. W.; Park, C. S. TDM-PON-based optical access network for tactile internet, 5G, and beyond. IEEE Netw. 2022, 36, 76–81.

[11] Kaneda, N.; Veen, D.; Mahadevan, A.; Houtsma, V. DSP for 50G/100G hybrid modulated TDM-PON. Proc. of ECOC 2020, 36, 1–2.

[15] Suzuki, N.; Miura, H.; Matsuda, K.; Matsumoto, R.; Motoshima, K. 100 Gb/s to 1 Tb/s based coherent passive optical network technology. J. Lightw. Technol. 2018, 36, 1485–1491.

[20] Kodama, T.; Nakagawa, T.; Matsumoto, R. Any-Double-Link Failure Tolerant Bypass/Backup Switchable WDM-PON Em-ploying Path-Pair Shared Protection and Bidirectional Wavelength Pre-assignment. Proc. of OFC 2022, W3G.5.

  1. On page 2, line 62, why the sentence starts with a full stop?

[Response] We appreciate the reviewer to point out the mistake. We have removed the full stop in the revision. Thanks again.

  1. On page-3, line 101, where is plot in the figure 2?

[Response] Thanks to the reviewer’s reminder. We have fixed the related subscripts to avoid confusion according to the reviewer’s advice in the revision. Please see the revised manuscript.

  1. On page 2-3, figure 3 must be cited close to where discussed in literature.

[Response] We have modified according to the reviewer’s suggestion. Thanks a lot.

  1. Why is figure 3(b) mentioned later in the manuscript in line 210 page 6?

[Response] This is because we want to clarify whether the power budget value of the experiment can meet the designed PON architecture, so it will be explained again here.

  1. One Page 6, Line 195-196, What does mean by "All the measured eyes can be distinguished clearly". Are you reffering to the legend in figure 6, clarify that.

[Response] We have modified the statement according to the reviewer’s suggestion in the revision.

The measured eyes of insets (ii) and (iv) are open and clear. And the insets (i) and (iii) also can be distinguished for observation.

  1. Clarify your approach with a suitable application.

[Response] We appreciate the reviewer’s valuable comment. We have included in the revision on page 5.

Therefore, the presented WDM-PON architecture can not only provide the fiber and FSO traffics simultaneously, but also can support the fiber fault protection in the fiber PON connection to provide the seamless signal access.

  1. Also include list of abbreviations at the end.

[Response] We would like to thank the reviewer’s useful comment. We have included the list of abbreviations in the end of the revision.

Abbreviations

The following abbreviations are used in this manuscript:

AP

Access Point

AWG

Arrayed Waveguide Grating

BER

Bit Error Rate

CO

Central Office

CP

Optical Coupler

EPON

Ethernet-PON

FEC

Forward Error Correction

FSO

Free Space Optics

GPON

Gigabit-PON

MZM

Mach-Zehnder Modulator

NG-PON

Next Generation-PON

OC

Optical Circulator

OND

Optical Distribution Network

ONU

Optical Network Unit

OOK

On-Off Keying

OWU

Optical Wireless Unit

PC

Polarization Controller

PD

Photodiode

PON

Passive Optical Network

PTP

Point to Point

PTMP

Point to Multi-Point

RN

Remote Node

SMF

Single-Mode Fiber

SW

Optical Switch

TDM

Time-Division-Multiplexing

TLS

Tunable Laser Source

WCP

C/L Band WDM Coupler

WDM

Wavelength-Division-Multiplexing

  1. You also have to compare it with a reported literature.

[Response] We have added in the revision according to the reviewer’s suggestion. Thanks a lot.

Compared with the related recent study [24], the proposed self-protected WDM-PON architecture can not only support the fiber and FSO transmissions at the same time, but also provide self-healing fiber breakpoint protection with simple and cost-effective implement.

  1. Make a table to indicate all the results in a comparative manner.

[Response] We would like to thank the reviewer’s valuable advice. We have included the measured results in Tab. 1 in the revision.

In addition, to clearly express the signal performance obtained in the PON system, the corresponding results of fiber and FSO signal is listed in Tab. 1.

Table 1. The corresponding results of fiber and FSO signal performances.

  1. Also the abstract is weak, revise it properly if possible.

[Response] We have modified the abstract according to the reviewer’s suggestion.

In this study, we propose a new wavelength-division-multiplexing passive optical network (WDM-PON) system to support the blended fiber-free space optics (FSO) signal access. To provide the fiber and FSO traffics simultaneously, the C- and L-band channels are applied in the presented PON, respectively. Moreover, to avoid the fiber breakpoint in the fiber access traffic, the proposed WDM access architecture also can provide the self-restored mechanism by applying simple fiber routing path. In addition, the corresponding signal performances of fiber and FSO channels are also executed experimentally for demonstration.

Reviewer 2 Report

the paper needs major revisions. please see the attached reviewer report

Author Response

Reviewer #2:

Examination of the manuscript shows that the present investigation is not good in the present form. but, the following major points should be addressed carefully during the revision to change my decision in the revision:

- Add a nomenclature table with SI units and all used abbreviations.

[Response] We have added and included the related table and abbreviations in the revision according to the reviewer’s suggestion as below.

In addition, to clearly express the signal performance obtained in the PON system, the corresponding results of fiber and FSO signal is listed in Tab. 1.

Table 1. The corresponding results of fiber and FSO signal performances.

Abbreviations

The following abbreviations are used in this manuscript:

AP

Access Point

AWG

Arrayed Waveguide Grating

BER

Bit Error Rate

CO

Central Office

CP

Optical Coupler

EPON

Ethernet-PON

FEC

Forward Error Correction

FSO

Free Space Optics

GPON

Gigabit-PON

MZM

Mach-Zehnder Modulator

NG-PON

Next Generation-PON

OC

Optical Circulator

OND

Optical Distribution Network

ONU

Optical Network Unit

OOK

On-Off Keying

OWU

Optical Wireless Unit

PC

Polarization Controller

PD

Photodiode

PON

Passive Optical Network

PTP

Point to Point

PTMP

Point to Multi-Point

RN

Remote Node

SMF

Single-Mode Fiber

SW

Optical Switch

TDM

Time-Division-Multiplexing

TLS

Tunable Laser Source

WCP

C/L Band WDM Coupler

WDM

Wavelength-Division-Multiplexing

- Respect the guidelines of the journal and its citation style.

[Response] We appreciate the reviewer’s kind reminder. We have followed the journal’s guidelines. Please refer to the revision.

- Provide a suitable reference for each used equation or model.

[Response] We have cited the appropriate references in the revised manuscript according to the reviewer’s suggestion. Please refer to the revision. Thanks a lot.

- The main findings should be highlighted in the abstract.

[Response] We have modified the abstract according to the reviewer’s comment as below.

In this study, we propose a new wavelength-division-multiplexing passive optical network (WDM-PON) system to support the blended fiber-free space optics (FSO) signal access. To provide the fiber and FSO traffics simultaneously, the C- and L-band channels are applied in the presented PON, respectively. Moreover, to avoid the fiber breakpoint in the fiber access traffic, the proposed WDM access architecture also can provide the self-restored mechanism by applying simple fiber routing path. In addition, the corresponding signal performances of fiber and FSO channels are also executed experimentally for demonstration.

- The main objectives of the study should be itemized at the end of the introduction.

[Response] We have stated the description according to the reviewer’s suggestion as below.

Compared with the related recent study [24], the proposed self-protected WDM-PON architecture can not only support the fiber and FSO transmissions at the same time, but also provide self-healing fiber breakpoint protection with simple and cost-effective implement.

- The authors have invited to incorporate real images for the realized experiences.

[Response] We have contained the real images in the experiments of Fig. 5 according to the reviewer’s comment. Thanks a lot.

Figure 5. Experimental setup of (a) fiber and FSO transmissions, respectively. The photos of insets are the corresponding active and passive components.

- Correlate the main graphical results by an accurate relationship.

[Response] Thanks to the reviewer’s reminder. We have modified according to the suggestion of the reviewer. Please refer to the revision. Thanks again.

- Improve the discussion more.

[Response] We have improved the discussion according to the reviewer’s advice in the revised manuscript. Please refer to the revision.

- Link the title with the abstract and conclusions.

[Response] Thanks to the reviewer providing the useful comment. We have modified the link in the revision.

- Remove all typos and grammatical errors.

[Response] We have corrected and modified the typos and errors according to the reviewer’s suggestion. Thanks a lot.

- Based on your results, how can the investigators increase the quality of these performance parameters?

[Response] To improve the signal performances of fiber and FSO traffics, the simplified PON architecture can improve the power budget for providing a longer optical fiber and FSO transmission distance. The statement has included in the revision according to the reviewer’s suggestion. Thanks a lot.

- the paper language should be revised carefully.

[Response] We have modified the revised manuscript according to the reviewer’s suggestion. Please refer to the revision. Thanks a lot.

The paper can not be accepted in its current form.

Reviewer 3 Report

Review of the manuscript Electronics-1780071

Integrated Fiber-FSO WDM Access System with Fiber Fault Protection

By Chin-Hung Yeh, Wen-Piao Lin, Sy-Yang Jiang, Shang-En  Hsieh, Ching-Huan Hsu and Chi-Wai Chow

The authors proposed a wavelength-division-multiplexing passive optical network (WDM-PON) system which simultaneously supports the blended fibre-free space optical communication (FSO) signal access. They also implemented the proposed system and investigated it experimentally. They used the C-band and L-band wavelengths of 1535.04 t0 1540.56 nm both in the optical fibre communication systems and in FSO. The paper is interesting for the researchers and engineers   occupied in optical communications and optical signal processing. The proposed architecture is novel as compared to the ones mentioned in references. The paper may be accepted for publication with the following minor revisions.

1.      Section 2, page 2, line 64. The authors should briefly describe “the arrayed waveguide grating (AWG)” and present essential characteristics of these optical waveguides such as material, optical parameters, dimensions.

2.      Section 3, page 5, line152. The authors used in the experiment “a C-band tunable laser source (TLS)”. Please, clarify what type of a laser was used in the experiment? It would be helpful to present the essential characteristics of TLS.

3.      The authors should present the typical values of the optical carrier power used in the experiment. Please, present the optical power limits typical for the proposed system.

4.      I recommend a brief discussion of the influence of nonlinear optical phenomena typical for optical fibre communication systems such as the stimulated Brillouin scattering (SBS), stimulated Raman scattering (SRS) on the performance of the proposed system

Author Response

Reviewer #3:

The authors proposed a wavelength-division-multiplexing passive optical network (WDM-PON) system which simultaneously supports the blended fibre-free space optical communication (FSO) signal access. They also implemented the proposed system and investigated it experimentally. They used the C-band and L-band wavelengths of 1535.04 t0 1540.56 nm both in the optical fibre communication systems and in FSO. The paper is interesting for the researchers and engineers   occupied in optical communications and optical signal processing. The proposed architecture is novel as compared to the ones mentioned in references. The paper may be accepted for publication with the following minor revisions.

  1. Section 2, page 2, line 64. The authors should briefly describe “the arrayed waveguide grating (AWG)” and present essential characteristics of these optical waveguides such as material, optical parameters, dimensions.

[Response] We would like to thank the reviewer’s suggestion. We have stated and included the optical characteristic of AWG on different pages in the revision as below.

Each WDM transceiver (TRx) is placed in the central office (CO) to transmit the corresponding downstream traffic and detect the upstream signal through the 1´N arrayed waveguide grating (AWG) device for WDM access, respectively. Each connected port of AWG has 3 dB bandwidth of 0.4 nm.

As we know, due to the periodic characteristic of AWG, the downstream wavelengths of l1C and l1L, l2C and l2L, … and lNC and lNL can enter the connected port of “1”, “2”, … and “N”, respectively. Hence, the downstream fiber and FSO signals of l1C and l1L, l2C and l2L, … and lNC and lNL are sent from the CO via the AWG1, AWG2, AWG3, and AWG4, respectively, and arrive the corresponding ONU1, ONU2, … and ONUN. Similarly, the upstream fiber and FSO signals of l(N+1)C and l(N+1)L, l(N+2)C and l(N+2)L, … and l(2N)C and l(2N)L are also through the same connected ports for data transmission via the AWGs, respectively, as seen in Fig. 2.

Based on the presented PON architecture of Fig. 2 and experimental setup of Fig. 5(a), the downstream signal will pass through the AWG3 (6 dB), SW (1 dB), 25 km fiber (0.2 ´ 25 = 5 dB) and OC2 (0.5 dB) after leaving the CO1 in normal state, respectively.

  1. Section 3, page 5, line152. The authors used in the experiment “a C-band tunable laser source (TLS)”. Please, clarify what type of a laser was used in the experiment? It would be helpful to present the essential characteristics of TLS.

[Response] Thanks to the reviewer’s comment. We have involved the description in the revision as below.

The TLS (General Photonics, TLS-101-C) has ITU-T specification tuning range from 1528.77 to 1563.86 nm with 13 dBm largest output power and 0.4 nm tuning step.

  1. The authors should present the typical values of the optical carrier power used in the experiment. Please, present the optical power limits typical for the proposed system.

[Response] We would like to thank the reviewer’s advice. We have stated in the original manuscript.

In the experiment, the output power of TLS at the port “2” of OC1 and OC2 is set at 7.5 dBm to carry the OOK modulation signal to avoid the nonlinear effect, as illustrated in Fig. 5(a).

  1. I recommend a brief discussion of the influence of nonlinear optical phenomena typical for optical fibre communication systems such as the stimulated Brillouin scattering (SBS), stimulated Raman scattering (SRS) on the performance of the proposed system

[Response] Thanks to the reviewer’s useful comment. We have stated the description in the discussion.

Due to the short transmission distance and limited power in the proposed PON system, the stimulated Brillouin scattering (SBS) and stimulated Raman scattering (SRS) can be ignored.

Round 2

Reviewer 2 Report

Authors replied effectively to all comments

Then the paper can be accepted